



# Assessment of neutrons from secondary cosmic rays at mountain altitudes – Geant4 simulations of environmental parameters including soil moisture and snow cover

Thomas Brall[1], Vladimir Mares[1], Rolf Bütikofer[2], and Werner Rühm[1]

[1]Helmholtz Zentrum München, Institute of Radiation Medicine, Ingolstädter Landstr. 1, 85764 Neuherberg, Germany
[2]University of Bern, Space Research & Planetary Sciences, Sidlerstrasse 5, 3012 Bern, Switzerland

**Correspondence:** Thomas Brall (tbrall@web.de)

**Abstract.** Ground based measurements of neutrons from secondary cosmic rays are affected by environmental parameters, particularly hydrogen content in soil. To investigate the impact of these parameters, Geant4 Monte Carlo simulations were carried out. In a previous study the model used for the Geant4 Monte Carlo simulations was already validated by measurements performed with an Extended Range Bonner Sphere Spectrometer (ERBSS) at Zugspitze, Germany, and at Jungfraujoch,

Switzerland. In the present study a sensitivity analysis including different environmental parameters (i.e., slope of mountain, snow height, soil moisture, and range of albedo neutrons) and their influence on the flux of neutrons from secondary cosmic rays was performed with Geant4. The results are compared with ERBSS measurements performed in 2018 at the Environmental Research Station "Schneefernerhaus" located at the Zugspitze, Germany. It is shown that the slope of the Zugspitze mountain reduces the neutron flux from secondary cosmic rays between about 25 % and 50 % as compared to a horizontal surface,

depending on neutron energy and snow cover. An increasing height of snow cover, simulated as snow water equivalent (SWE), reduces the total neutron flux exponentially down to a factor of about 2.5 as compared to soil without any snow cover, with a saturation for snow heights greater than 10 cm to 15 cm SWE, depending on neutron energy. Based on count rates measured with the individual spheres of the ERBSS, SWE values were deduced for the whole year 2018. Specifically, mean SWE values deduced for the winter months (January to March) are between 6.7 and 10.1 cm or more, while those for the summer months

(July to September) are between 2.1 and 3.6 cm. Soil moisture of 5 % water mass fraction in limestone leads to a decrease of the total neutron flux by about 35 % compared to dry limestone. At a height of 1.5 m above ground, 86 % of the total albedo neutron fluence at the detector position are from a ground area with a radius of about 75 m. It is concluded that measurement of neutrons from secondary cosmic radiation can be used to gain information on height of snow cover and its seasonal changes, soil moisture, but also information on local geometry such as mountain topography. Because the influence of such parameters

on neutron fluence from secondary cosmic rays depends on neutron energy, analysis of the whole neutron energy spectrum is beneficial.



## 1 Introduction

Neutrons from secondary cosmic rays (CRs) are always present at the Earth's surface as a component of natural radiation background. These neutrons are produced during cascade reactions in the Earth's atmosphere by primary CRs (mainly protons and helium nuclei). When reaching the atmosphere, the CR particles interact with the atoms of the air (basically oxygen and nitrogen atoms) and are continuously slowed down due to ionization. Additionally, they interact with the nuclei of these atoms and new particles like protons, neutrons, $\pi$ and K mesons are produced characterized by a wide spectrum of energies extending up to several GeV. Fast neutrons of energies below 10 MeV may be evaporated from excited target nuclei. When reaching the ground, the transport of fast neutrons through soil is strongly influenced by the presence of hydrogen, which has the ability to rapidly moderate neutrons due to its large elastic scattering cross section. Hydrogen at the land surface is mainly in the form of liquid and solid (ice, snow) water. This fact should be taken into account, if spectral fluence rate distributions of neutrons from secondary CRs are measured on the ground level. It has been observed that snow accumulation in the environment of neutron detectors has a significant effect on the measured neutron flux energy distribution, because it influences the intensity of the ground albedo neutron flux (e.g., Tanskanen, 1968; Eroshenko et al., 2008; Rühm et al., 2012).

It has long been recognized that the measurement of albeo neutrons can be used to detect environmental hydrogen. For example, it has been proposed that albedo neutrons produced by cosmic radiation on the surface of Mars (or Moon) could be used to detect the water content of Martian soil (e.g., Mitrofanov et al., 2004). Along these lines it has been also proposed to use neutrons from secondary cosmic radiation near the Earth's surface to detect soil moisture (Zreda et al., 2012; Andreasen et al., 2017). Specifically, it was shown that detectors measuring thermal and/or epithermal neutrons close to the soil surface provide a signal that depends on soil moisture, within a radius in the order of about hundred meters around the detector position (Köhli et al., 2015). A network of such neutron probes has already been installed for example in the US (Zreda et al., 2012) and the UK (Evans et al., 2016). Mobile systems were also proposed (Schrön et al., 2018).

In the beginning of the 20th century, after the discovery of the cosmic rays in 1912 by Victor Hess (Hess, 1912), mainly ionization chambers were used to measure the intensity of the cosmic rays. In the 1950's the neutron monitor (NM), developed by Simpson ?Meyer and Simpson, 1955), was considered as the best ground-based detector capable to record variations of primary CR intensity. Since the late 1950's, a global NM network [http://www.nmdb.eu] was built to record long- and short-term changes of the CR intensity at ground level. NMs are sensitive to secondary particles produced in atmospheric cascades (mainly secondary neutrons) from primary CRs. They use the neutron-induced nuclear reactions ((n,2n), (n,3n)) in lead included in their structure to multiply the number of secondary neutrons, which are then moderated to thermal energies and finally detected in the proportional counter tubes filled with $^{10}BF_3$ (or $^3He$) gas through the detection of charged particles produced for example by the $^{10}B(n,\alpha)^7Li$ (or $^3He(n,p)^3H$) reaction. This neutron multiplication technique increases significantly the counting rate of NMs and improves its statistical accuracy. Because the number of produced secondary neutrons is almost independent on the energy of the incident neutron, a single NM cannot be used as a neutron spectrometer. If the neutron fluence is required as a function of neutron energy in the range from thermal energies up to several GeV, an Extended-Range Bonner Sphere Spec-



trometer (ERBSS) has to be used (Schraube et al., 1997; Mares and Schraube, 1998), which is based on the initial standard
Bonner sphere spectrometer (BSS) (Bramblett et al., 1960).

The effect of hydrogen in snow on the flux spectra of secondary neutrons from CRs at ground level has recently been
demonstrated by measurements at mountain altitude at the Environmental Research Station (UFS) "Schneefernerhaus" located
at the Zugspitze mountain, Germany, and at sea level at the Koldewey station on Spitsbergen (Rühm et al., 2012). Specifically,

it was shown that the fluence of thermal and epithermal neutrons change by a factor of two between summer and winter season,
at the UFS, while they change by about 50% at Spitsbergen. Seasonal changes in fluence rate of MeV neutrons were roughly
a factor of two smaller, at both locations, while the fluence rate of 100 MeV neutrons did not change much at both locations.
These changes were qualitatively attributed to the presence of snow during winter times and the absence of snowduring summer
times, but a quantitative evaluation of such an effect is still missing.

In this paper, detailed Monte Carlo (MC) simulations are described which allow quantification of the influence of environ-
mental parameters such as snow cover and soil humidity on the energy spectrum of secondary CR neutrons. In a previous paper,
first simulations were validated by means of experimental spectrometry using the ERBSS on the UFS, and at Jungfraujoch,
Switzerland, performed during winter and summer seasons, respectively (Brall et al., 2021). While there was overall agreement
(within about 35%) between measured and simulated neutron fluences for energies above about 20 MeV, the comparison of

measured and simulated neutron flux spectra below 20 MeV was limited by the unknown hydrogen content in the environment
close to the measurement locations. In the study described in the present paper, sensitivity analyses were carried out to investi-
gate the influence of environmental parameters on simulated neutron flux spectra, with emphasis on the energy range between
thermal and MeV energies. Specifically, it was investigated whether variations in height of snow cover and soil humidity in
the environment of the measurement locations can improve the agreement between measurement and simulation at neutron

energies below 20 MeV.

## 2 Materials and Methods

### 2.1 Monte Carlo Simulations

#### 2.1.1 Overall Procedure

In a previous paper (Brall et al., 2021), MC simulations using the Geant4 toolkit were used to assess the albedo neutron flux

for two locations at mountain altitudes, one at the UFS on the Zugspitze mountain, Germany, the other at the High Altitude
Research Station Jungfraujoch, Switzerland. While the station at the Jungfraujoch is located at an altitude of 3,582 m a.s.l. on
the top of a steep hill named "Sphinx", the station at the Zugspitze mountain is located on the southern slope of the Zugspitze
mountain at an altitude of 2,660 m a.s.l.. The GEANT simulations described in Brall et al. (2021) were done including three
different physics lists "QGSP_BERT_HP", "QGSP_BIC_HP", and "Shielding", which are all reference physics lists of the

Geant4 toolkit (Geant4 Collaboration, 2017). For the present paper, these three physics lists were also used. Because the





results of the simulations do not show any substantial differences between the physics lists, however, all results are presented and discussed here as obtained using the QGSP_BERT_HP (Bertini) physics list.

In order to reduce CPU computation time, the simulations were performed in a two-step process. In the first step, the primary CR particles were started at an altitude of 100 km and propagated down to a selected altitude. There the momenta of
all secondary particles were scored in a pre-defined boundary surface. In the second step, the particles scored were then used as source particles and further propagated, to investigate the influence of local environmental parameters on neutron flux spectra at ground level.

### 2.1.2 Simulation Geometries

The geometry implemented in the Geant4 toolkit to simulate neutrons from secondary CR at mountain altitudes is described
in detail in Brall et al. (2021). Briefly, the primary radiation source includes protons and alpha particles from primary CR impinging on the top of the atmosphere (Burger et al., 2000; Usoskin et al., 2005). The parameters of the Earth's atmosphere and its elemental composition are implemented according to the US Standard Atmosphere 1976 [COESA 1976]. For the present study, a horizontal flat soil disk was used to investigate the influence of environmental parameters such as height of snow cover and soil moisture. A layer of limestone ($CaCO_3$) with a thickness of 10 m and a density of 2.7 $g/cm^3$ was chosen as soil
material, because limestone is typical for the rock material in the investigated alpine region of the Zugspitze mountain.

a) Height of snow cover on dry limestone

As a boundary surface in the first step of the simulation, a disc with a radius of 1,000 m is chosen at an altitude of 2,700 m as the source. The radius of the soil is 20 km. The scorer is at an altitude of 2,651.5 m, i.e., 1.5 m above the soil level, and consists of a disc with a radius of 500 m. On top of the soil, layers of water with various thickness are placed (1, 2, 5, 10, 12.5, 20, and
50 cm snow water equivalent (SWE)) roughly simulating a snow layer with a thickness between 0 cm and 200 cm (Fig. 1). In all simulations, a snow density of 250 $kg/m^3$ was assumed which had been measured in the vicinity of the UFS (Hürkamp et al., 2019) . Note that typically, snow densities range from 50 $kg/m^3$ (New Snow) to 800 $kg/m^3$ (Very wet snow and firn) (Cuffey and Paterson, 2010).

b) Soil moisture

In order to investigate the influence of soil moisture on neutron flux spectra of secondary CRs, water was added in the limestone in various mass fractions (0, 0.5, 1, 2, and 5 %). Such simulations were performed without any snow cover (i.e., 0 cm SWE) and with a snow cover (i.e., 5 cm SWE) (Fig. 1).

c) Effect contribution of albedo neutrons as a function of distance In order to investigate from which distance albedo neutrons contribute to the neutron energy spectrum at a given location (i.e., at detector position), a scorer with a radius of 5 m
was placed 1.5 m above the limestone soil including 40 cm of snow on top (simulated with water of 250 $kg/m^3$ density, 10 cm SWE). For this study, the soil was comprised of a 10 m thick disk with various radii (0, 5, 10, 15, 20, 30, 40, 50, 60, 70, 80, 90, 120, 150 m). Outside the disk vacuum was assumed. The source was assumed at an altitude of 2,700 m with a radius of 320 m (Fig. 2).



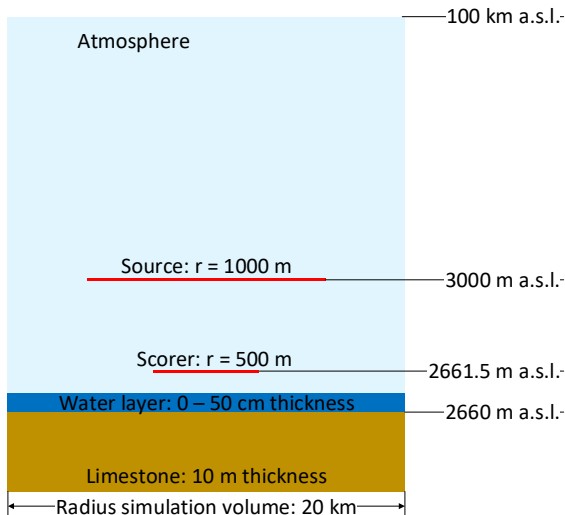

**Figure 1.** Geometry implemented in Geant4 to evaluate the effect of water layers of different heights above the limestone

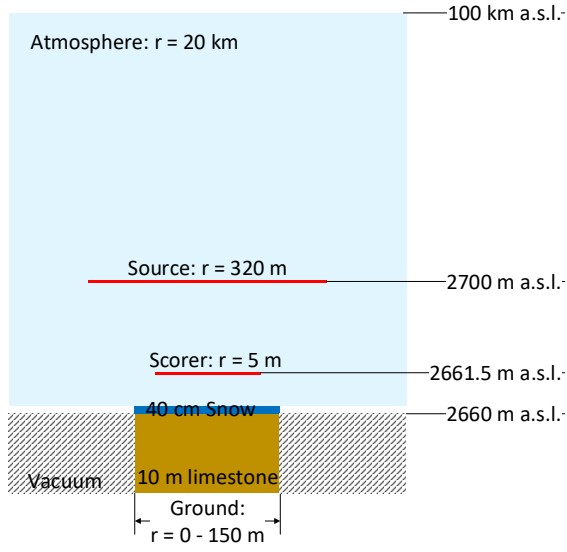

**Figure 2.** Geometry implemented in Geant4 to evaluate the effect of the size of the snow layer area on the albedo neutron flux at the scorer location.

## 2.2 Measurement of neutrons from secondary cosmic radiation

At the Zugspitze mountain, Germany, the neutron flux spectra from secondary CRs was measured using an ERBSS. The spectrometer is described in detail in (Schraube et al., 1997; Mares and Schraube, 1998; Leuthold et al., 2007; Rühm et





al., 2008; Mares et al., 2020; Brall et al., 2020, 2021). Briefly, 16 $^3$He proportional counters were simultaneously used to detect thermalized neutrons. Thermalisation was achieved by covering the counters with polyethylene (PE) spheres of various

thicknesses (except for one proportional counter which was not covered by any PE, to detect environmental neutrons already thermalized). The response functions for the various counters were calculated for the energy range from meV up to GeV with MC simulations (Mares et al., 1991; Mares and Schraube, 1998). Count rates of the proportional counters together with their response functions were finally unfolded to obtain flux distributions as a function of neutron energy at the spectrometer positions.

## 130   3   Results and Discussion

### 3.1   Influence of Environmental Parameters

#### 3.1.1   Difference between horizontal and slanted soil

In a previous paper (Brall et al., 2021), results of measurements of neutron flux spectra at the slope of the Zugspitze mountain are presented. In this work, simulations of the neutron flux spectra for the measurement location were done using a similar

approach as was used for the present study. Consequently, the neutron flux spectra simulated in the previous paper for a slant angle of 45° (Brall et al., 2021) could be compared with those obtained in the present study using horizontal ground. This was done for both dry limestone without snow cover, and dry limestone with a cover of 50 cm of snow corresponding to 12.5 cm SWE.

     Table 1 shows the ratio of the fluence between horizontal geometry and slanted geometry, for four energy regions (thermal:

$E < 0.4$ eV; epithermal: $0.4$ eV $\leq E < 100$ keV; evaporation: $100$ keV $\leq E < 20$ MeV; cascade: $E \geq 20$ MeV). Table 1 demonstrates that for the horizontal geometry, the fluence of high-energy cascade neutrons ($E \geq 20$ MeV) is higher by some 25% as compared to slanted soil, probably due to the fact that part of the $2\pi$ geometry of the sky (i.e., upper half sphere) is shielded by the slanted surface. In contrast, for lower-energy neutrons, and in particular for thermal and epithermal neutrons, this effect increases by about 50%. This can be explained by the mainly downward direction of cascade neutrons, while the

lower-energy neutron flux is more isotropically distributed. This effect is roughly similar whether or not an additional water layer on top of the ground is considered. The corresponding neutron flux spectra are shown in Fig. 3.

#### 3.1.2   Influence of snow height on neutron flux spectra

Simulations

As already mentioned earlier, seasonal changes in the neutron flux spectra had been measured at the UFS (Rühm et al., 2012).

In order to interpret the results of these measurements, the influence of snow height on the neutron flux spectrum was simulated in the horizontal geometry, and the results are shown in Fig. 4. In Fig. 5, the corresponding total flux and the flux for the four energy regions (thermal: $E < 0.4$ eV; epithermal: $0.4$ eV $\leq E < 100$ keV; evaporation: $100$ keV $\leq E < 20$ MeV; cascade: $E \geq 20$ MeV) are shown as a function of the thickness of a water layer covering the ground.




| Snow Water Equivalent (SWE) | 0cm | 12.5 cm |
|---|---|---|
| Total | 1.44 | 1.34 |
| Thermal (E< 0.4 eV) | 1.49 | 1.36 |
| Epithermal (0.4 eV $\leq$ E < 100 keV) | 1.52 | 1.46 |
| Evaporation (100 keV $\leq$ E < 20 MeV) | 1.40 | 1.38 |
| Cascade (E $\geq$ 20 MeV) | 1.25 | 1.23 |

**Table 1.** Ratio of neutron fluence for horizontal geometry to that for slanted geometry (slant angle 45°); thermal: E < 0.4 eV; epithermal: 0.4 eV $\leq$ E < 100 keV; evaporation: 100 keV $\leq$ E < 20 MeV; cascade: E $\geq$ 20 MeV.

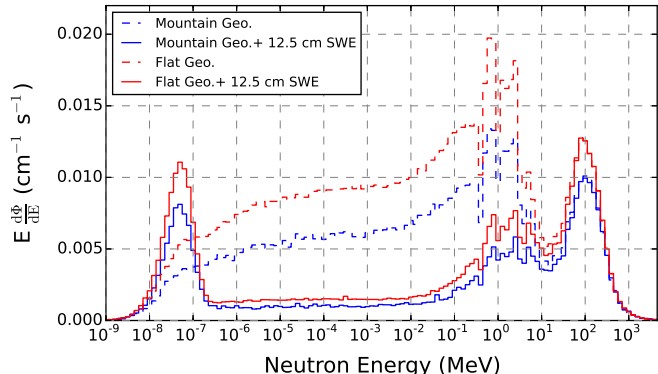

**Figure 3.** Top: Neutron flux spectra for horizontal (red) and slanted soil (blue). With 50 cm snow layer (corresponding to 12.5 cm SWE) (solid line) and without snow (dashed line).

For thermal neutrons, the fluence rate increases with increasing thickness of the water layer up to a thickness of about 3 cm, due to thermalization of higher-energy neutrons by the hydrogen in water. Beyond a thickness of about 3 cm, the neutron flux decreases again, due to absorption of neutrons backscattered from soil by the overlying water layer. In contrast, neutrons with higher than thermal energies do not show such an initial increase with increasing water thickness, but decrease from the beginning almost exponentially with increasing water thickness, due to neutron moderation (Fig. 5). This decrease is most prominent for epithermal neutrons, followed by evaporation neutrons. In contrast, for high-energy cascade neutrons, this decrease is small and amounts only to about a few percent, probably because only few of these neutrons are backscattered by the soil and get absorbed or moderated by the overlying water layer. Interestingly, for any of the investigated energy regions, the neutron flux saturates at about 20 cm thickness of water layer (Fig. 4 and Fig. 5) and changes only little for greater water thickness. This means that the neutron energy spectrum does not change its shape substantially for water thicknesses greater than about 20 cm (corresponding to snow heights greater than about 60-80 cm, depending on snow density).

Based on Fig. 6 the mean minimum and maximum values of the neutron flux in winter (Jan – March and December 2018) and summer (Jul – Oct 2018) were calculated and, as a measure of the amplitude of the observed changes, the difference

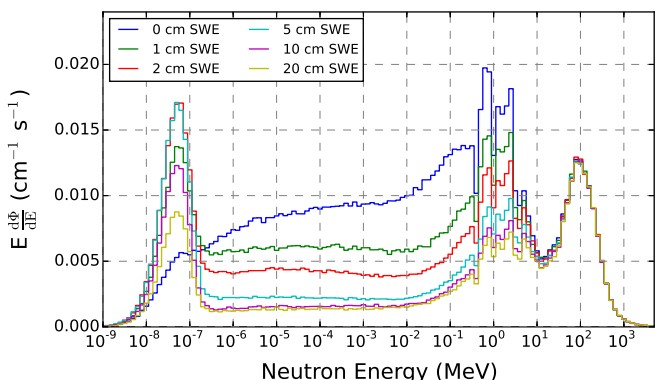

**Figure 4.** Neutron flux spectra simulated with water layers of different heights on ground.

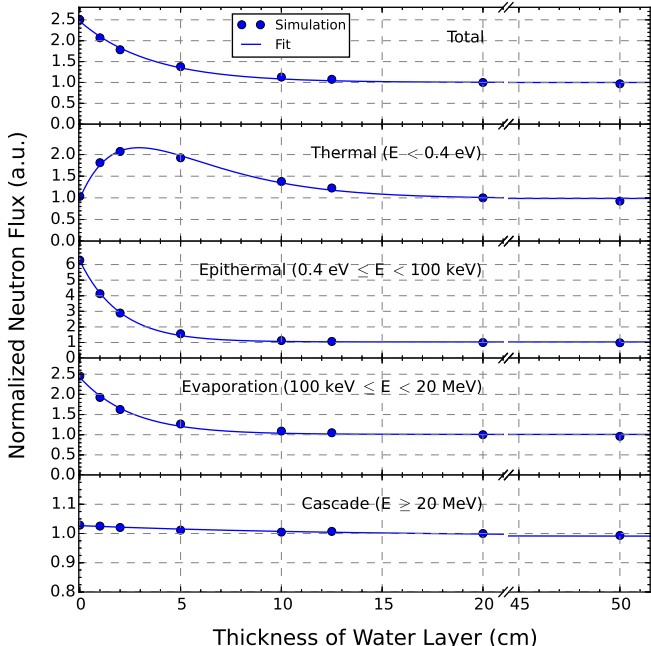

**Figure 5.** Neutron flux as a function of thickness of water layer on dry limestone soil at 2,661.5 m a.s.l., for horizontal geometry and for different neutron energy ranges (thermal: E < 0.4 eV; epithermal: 0.4 eV ≤ E < 100 keV; evaporation: 100 keV ≤ E < 20 MeV; cascade: E ≥ 20 MeV; and total). Solid lines are representing fits of the data points using a function of $a\,e^{-bx} + c$ for epithermal, evaporation, and cascade neutrons, and $a\,x\,e^{-bx} + c$ for thermal neutrons.

between maximum and minimum values was divided by 2. For the four investigated energy regions, the results fit reasonably well with those reported in Rühm et al. (2012) where the corresponding amplitudes were obtained from a sinus function fitted over a period of 3 years (Table 2).





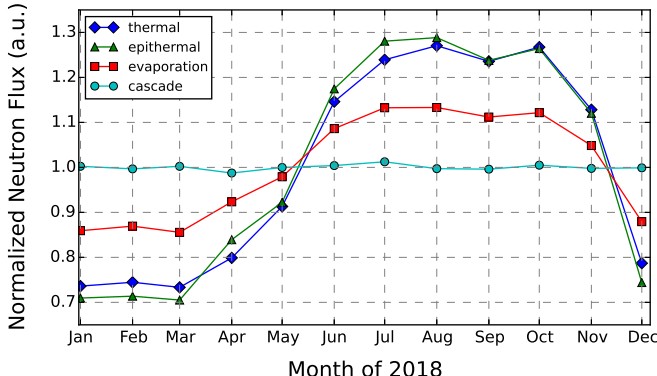

**Figure 6.** Monthly averaged neutron flux in 2018 at UFS Zugspitze, deduced from ERBSS neutron flux spectra for 4 energy regions (thermal: $E < 0.4$ eV; epithermal: $0.4$ eV $\leq E < 100$ keV; evaporation: $100$ keV $\leq E < 20$ MeV; cascade: $E \geq 20$ MeV) normalized to annual mean flux.

| | mean min | mean max | max/min | (max-min)/2 | Amplitude (Rühm et al., 2012) |
|---|---|---|---|---|---|
| Thermal | $0.75 \pm 0.02$ | $1.25 \pm 0.02$ | $1.67 \pm 0.03$ | $0.25 \pm 0.01$ | $0.27 \pm 0.02$ |
| Epithermal | $0.72 \pm 0.02$ | $1.27 \pm 0.02$ | $1.77 \pm 0.03$ | $0.28 \pm 0.01$ | $0.29 \pm 0.02$ |
| Evaporation | $0.87 \pm 0.01$ | $1.13 \pm 0.01$ | $1.30 \pm 0.010$ | $0.13 \pm 0.01$ | $0.14 \pm 0.01$ |
| Cascade | $1.0001 \pm 0.002$ | $1.003 \pm 0.007$ | $1.003 \pm 0.007$ | $0.001 \pm 0.004$ | $0.017 \pm 0.005$ |

**Table 2.** Normalized mean minimum (for Jan-March and Dec 2018) and maximum (for Jul – Oct 2018) neutron flux values (incl. standard deviation) taken from Fig. 6, difference of these values divided by 2, and corresponding data taken from Rühm et al. (2012).

Based on the ratios between maxima and minima ($1.67 \pm 0.03$ for thermal, $1.77 \pm 0.03$ for epithermal, $1.30 \pm 0.01$ for evaporation neutrons), snow water equivalent values can be estimated for the summer months (July – October 2018) based on the fits shown in Fig. 5. As a result, for thermal neutrons SWE values of about 1 cm or 6 cm can be deduced. In contrast, for epithermal neutrons an unambiguous SWE value of about 4 cm and for cascade neutrons 6 cm can be deduced.

    Because the fluence of the cascade neutrons does not change much with season, an analysis similar to that for thermal, 175   epithermal and evaporation neutrons as given above was not considered reasonable for cascade neutrons.

    The results obtained for thermal, epithermal and evaporation neutrons suggest (if the neutron flux saturated during the winter months (for SWE values greater than about 15 cm; see Fig. 5)) that during the summer months there was a mean snow water equivalent of about 4-6 cm in the vicinity of the ERBSS.

    This analysis already suggests that it is possible to deduce the height of snow cover from analysis of the ERBSS neutron 180   flux spectra. This encouraged us to perform a more detailed analysis, which is based on the pure count rates of the ERBSS proportional counters and which avoids the use of the unfolding process and associated uncertainties. This approach is described below.





Interpretation in terms of thickness of water layer

In order to validate these results with experimental data, measurements that had been made at the UFS, which is located at the slanted slope of the Zugspitze mountain at an altitude of 2,660 m a.s.l., were used. Specifically, we used the count rates obtained by the $^3$He proportional counters of the ERBSS for the comparison.

For this purpose, in a first step we used the simulated Geant4 neutron flux spectra obtained for the dry horizontal ground without overlying water layer (see above). Because the measurements had been done on the slanted slope of the Zugspitze

mountain (slant angle: 45°), the neutron flux spectra were corrected for the ratio of the simulated spectra of slanted to flat geometry (Table 1). In that way, a simulated neutron energy spectrum for the slanted slope was constructed. This spectrum was then folded with the response functions of the 18 proportional counters of the ERBSS, to calculate the count rates of the $^3$He proportional counters of the ERBSS. In other words, these count rates would be the count rates expected from an ERBSS based on the Geant4 simulations, for a slanted slope and ideal dry conditions (i.e., no water in soil, no water on top of the

soil, and no water in all the structures surrounding the ERBSS (e.g., building material, water pipes, etc.)). This procedure was also applied to the Geant4 neutron flux spectra simulated in the present work for dry ground with overlying water layers with various thickness. That is, the count rates were again calculated (by folding with the corresponding response functions) that are expected for an ERBSS located on a slanted slope including overlying water layers with various thickness. In a second step, these count rates were plotted as a function of water layer thickness, for each of the involved proportional counters, and fitted

with the function $a\,e^{-bx} + c$ or $a\,x\,e^{-bx} + c$ for the bare detector, where x is the thickness of the corresponding water layer. The result is shown in Fig. 7 , and the corresponding fit parameters are listed in Table 3.

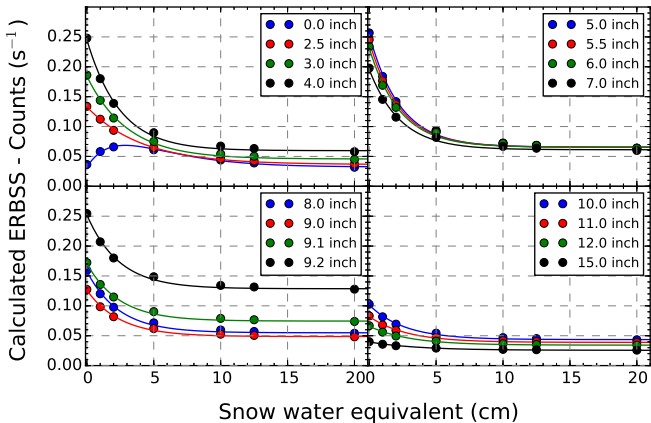

**Figure 7.** Count rates calculated for a solar modulation of 420 MV, for the $^3$He proportional counters of the used Extended-Range Bonner Sphere Spectrometer (ERBSS), based on simulated Geant4 neutron flux spectra and response functions of the ERBSS. Solid lines are the fit of the data points to $a\,e^{-bx} + c$ or $a\,x\,e^{-bx} + c$ for the bare detector.





| Sphere | a | b | c |
|--------|---|---|---|
| Bare | $0.034 \pm 0.003$ | $0.35 \pm 0.02$ | $0.032 \pm 0.001$ |
| 2.5" | $0.097 \pm 0.001$ | $0.25 \pm 0.01$ | $0.036 \pm 0.001$ |
| 3" | $0.139 \pm 0.003$ | $0.33 \pm 0.02$ | $0.046 \pm 0.002$ |
| 4" | $0.186 \pm 0.005$ | $0.41 \pm 0.03$ | $0.060 \pm 0.002$ |
| 5" | $0.189 \pm 0.005$ | $0.44 \pm 0.03$ | $0.065 \pm 0.002$ |
| 5.5" | $0.177 \pm 0.005$ | $0.44 \pm 0.03$ | $0.066 \pm 0.002$ |
| 6" | $0.166 \pm 0.005$ | $0.44 \pm 0.03$ | $0.066 \pm 0.002$ |
| 7" | $0.135 \pm 0.004$ | $0.43 \pm 0.03$ | $0.061 \pm 0.002$ |
| 8" | $0.103 \pm 0.003$ | $0.42 \pm 0.03$ | $0.055 \pm 0.001$ |
| 9" | $0.077 \pm 0.002$ | $0.40 \pm 0.03$ | $0.049 \pm 0.001$ |
| 9,1" | $0.096 \pm 0.003$ | $0.42 \pm 0.03$ | $0.075 \pm 0.001$ |
| 9,2" | $0.124 \pm 0.004$ | $0.42 \pm 0.03$ | $0.129 \pm 0.002$ |
| 10" | $0.058 \pm 0.002$ | $0.38 \pm 0.03$ | $0.044 \pm 0.001$ |
| 11" | $0.043 \pm 0.002$ | $0.36 \pm 0.03$ | $0.039 \pm 0.001$ |
| 12" | $0.032 \pm 0.001$ | $0.34 \pm 0.03$ | $0.034 \pm 0.001$ |
| 15" | $0.014 \pm 0.001$ | $0.27 \pm 0.03$ | $0.0256 \pm 0.0003$ |

**Table 3.** Fit Parameters of the fitted functions ($a\,e^{-bx} + c$ or $a\,x\,e^{-bx} + c$ for the bare detector) shown as solid lines in Fig. 7.

The daily ERBSS count rates actually measured in 2018 at UFS are shown in Fig. 8. These count rates were used to calculate, for every sphere and every day in 2018, the corresponding water thickness using the exponential function and the corresponding parameters from Table 3. Fig. 9 shows these resulting water thicknesses for 2018 at UFS based on the count
rates of the 16 ERBSS counters. Although the larger spheres and those including lead shells are less sensitive to low-energy (thermal, epithermal) neutrons than the smaller spheres, the overall annual count rates show qualitatively similar patterns, also taking into account the involved statistical uncertainites. Corresponding unfolded neutron flux spectra are shown in Fig. 10.

From the daily water thickness values for the winter months (January until March) and the summer months (July until September), mean values were then calculated for each sphere (Table 4). For the winter months, all spheres give reasonable
results between 6.7 and 10.1 cm SWE. The values from the 9.2" sphere (9" PE sphere with lead shell) and the 11" sphere are considerably lower than the values computed for the other spheres. For the summer months, water thicknesses between 2.1 and 3.6 cm SWE were obtained. Here, the thickness from the bare counter, and the thicknesses from the 2.5", 9,2" and 11" spheres are somewhat lower than the other values.

For the summer months when there is no snow at the UFS, the measured values are between 2.7 and 3.6 cm SWE. This
can be explained by the simplified assumptions used in the simulations. For example, complete dry limestone was assumed as a ground, and the building of the UFS research station was not considered. Thus, any contribution of the water content in the environment such as the typical 3-10 % water content of concrete, soil moisture and any additional water content in the





concrete floor around the detector housing was not considered in the simulations. For the winter months it can be seen from Fig. 8 that the count rates saturate and reach a minimum, corresponding to a maximum thickness of measurable SWE with
ERBSS.

These results indicate that during the summer months a minimum in SWE is obtained. SWE values can also be measured during the snow melting period in spring and the beginning of the snow fall period in autumn, because these months fall into periods where the count rates of the ERBSS proportional counters change and, thus, the deduced SWE thickness (Fig. 6). In contrast, the count rates of the ERBSS proportional counters saturate during winter times and, thus, SWE thicknesses of more
than 10 - 20 cm SWE cannot be determined based on ERBSS data (Fig. 6).

We note that the difference in SWE between winter and summer season is in most cases between 4 and 6 cm, see Table 4. This fits qualitatively to the results obtained when Fig. 5 and Table 2 are discussed (see above). It is also noted that the computations of the SWE based on measured ERBSS data implicitly include any humidity in the environment, for example the water content of the nearby concrete structure of the building and of the soil.

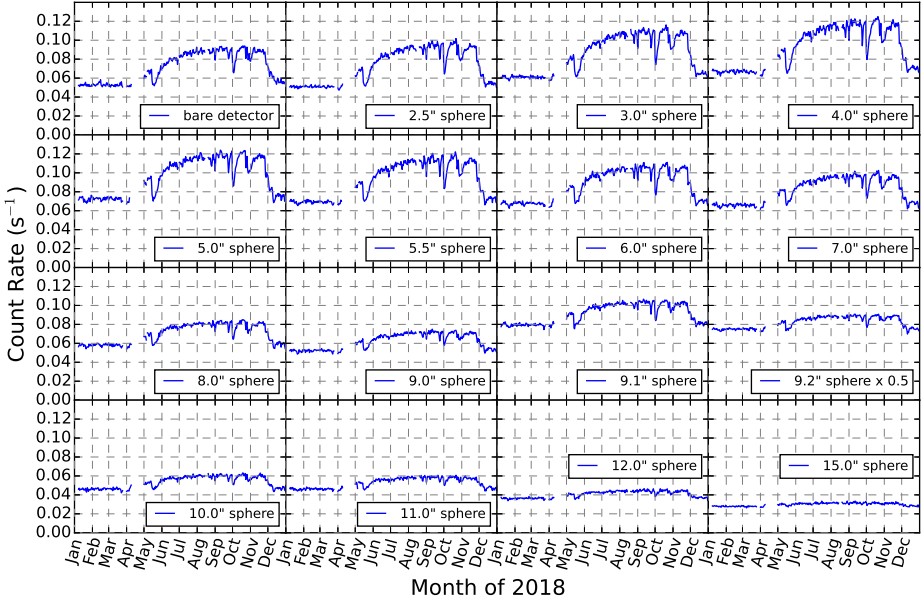

**Figure 8.** Measured daily count rates of the ERBSS proportional counters at UFS in 2018. Note that the count rates of the 9.2" sphere are divided by 2 to fit into the scale.

### 3.1.3 Fluence contribution of albedo neutrons as a function of distance

The influence of snow with a thickness corresponding to 10 cm SWE covering the soil around the scorer position at different distances is shown in Fig. 11. As mentioned in section 2, the soil and the water layer were assumed as finite discs with radii of 0, 5, 10, 15, 20, 30, 40, 50, 60, 70, 80, 90, 120 and 150 m for this study. The results for the neutron fluences of the different

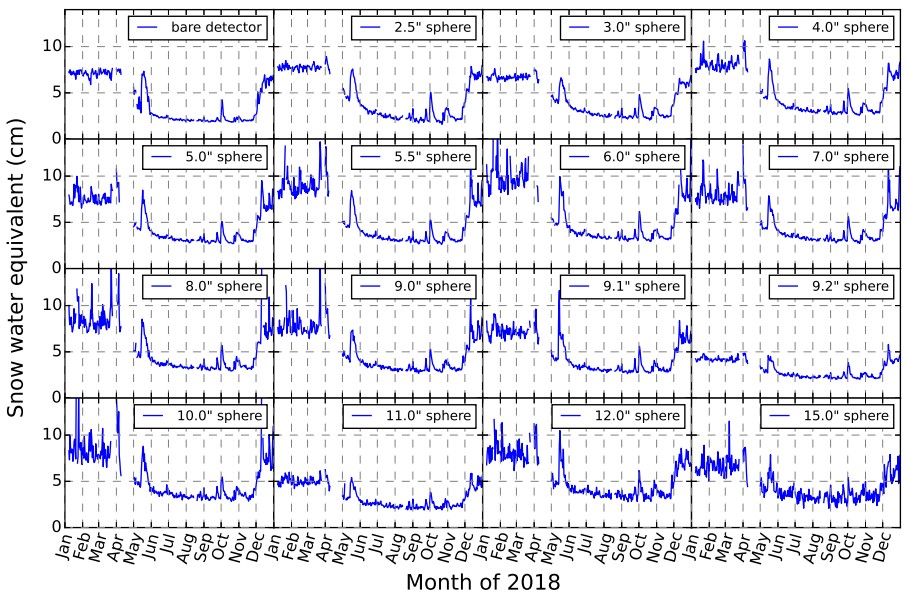

**Figure 9.** Daily calculated SWE in 2018 based on ERBSS count rates measured at UFS, the exponential fits shown in Fig. 7 and the corresponding fit parameters listed in Table 3. Thereby the QGSP BERT HP Geant4 physics lists were used to simulate the nuclear interactions.

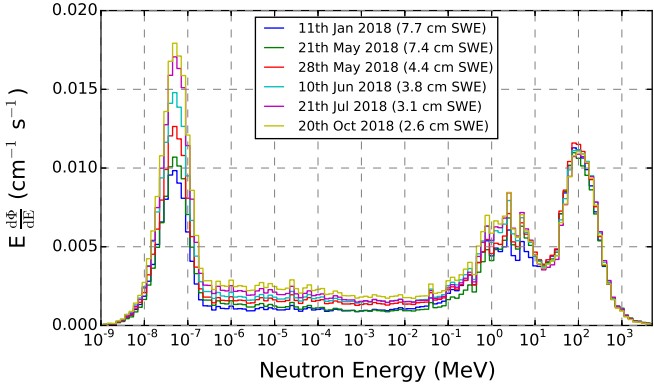

**Figure 10.** Unfolded ERBSS neutron flux spectra at UFS for selected dates in 2018; SWE calculated with the count rates of the 4" sphere.

neutron energy regions (thermal: E < 0.4 eV; epithermal: 0.4 eV ≤ E < 100 keV; evaporation: 100 keV ≤ E < 20 MeV; cascade: E ≥ 20 MeV) were fitted by a function of the form $a \arctan(bx)$ where x is the distance from the center of the soil disc (Fig. 2). For the total backscattered neutron fluence, 80 % come from distances of less than 40 m. For thermal and epithermal neutrons, 80 % come from distances of less than 50 m. In contrast, for evaporation and cascade neutrons, 80% come from distances of less than 25 m. We note that a smaller scorer than the one used here, i.e. a disc with radius 5 m, would reduce the area from where neutrons contribute to the detector signal.





| Sphere | SWE (cm) (Jan-Mar) | SWE (cm) (Jul-Sep) |
| --- | --- | --- |
| Bare | 7.11 ± 0.34 | 2.13 ± 0.35 |
| 2.5" | 7.77 ± 0.27 | 2.43 ± 0.56 |
| 3" | 6.71 ± 0.27 | 2.66 ± 0.44 |
| 4" | 8.04 ± 0.57 | 3.13 ± 0.46 |
| 5" | 7.60 ± 0.66 | 3.16 ± 0.42 |
| 5.5" | 9.03 ± 1.18 | 3.21 ± 0.45 |
| 6" | 10.07 ± 1.83 | 3.50 ± 0.52 |
| 7" | 8.04 ± 1.04 | 3.31 ± 0.47 |
| 8" | 8.58 ± 1.30 | 3.45 ± 0.46 |
| 9" | 7.88 ± 1.52 | 3.25 ± 0.43 |
| 9,1" | 7.24 ± 0.63 | 3.15 ± 0.47 |
| 9,2" | 4.21 ± 0.22 | 2.34 ± 0.30 |
| 10" | 8.36 ± 1.52 | 3.48 ± 0.46 |
| 11" | 5.00 ± 0.39 | 2.36 ± 0.35 |
| 12" | 8.13 ± 1.06 | 3.64 ± 0.54 |
| 15" | 6.68 ± 0.93 | 3.28 ± 0.57 |

**Table 4.** Snow water equivalent deduced from measured ERBSS count rates as calculated with the fit functions shown in Fig. 7 and the corresponding fit parameters listed in Table 3 . Shown are the mean values from the daily values of Fig. 9, for winter times (January - March) and summer times (July - September) and their standard deviations.

Köhli et al. (2015) found in their simulations that 86% of the neutron flux scored at a height of 2 m above ground originate from an area with a radius between 130 m and 240 m around the scorer position, depending on soil moisture, air humidity, and vegetation. This radius corresponds to a radius of about 75 m obtained in the present study (see Fig. 11). A quantitative comparison of our results with the results by Köhli et al. (2015) remains difficult, because of differences in the simulation methods, used geometry (e.g., height of scorer position), and different environmental parameters used such as, for example, air

pressure (due to different altitudes), soil composition, soil moisture, snow cover, and air humidity.

### 3.1.4    Humidity of Limestone

The influence of soil moisture on the neutron flux spectra is shown in Fig. 12. For both geometries (i.e., pure soil, and soil covered by a water layer with a thickness of 5 cm corresponding to a snow layer of about 20 cm), water mass fractions 0, 0.5, 1, 2 and 5 % (which were used as examples) were implemented in the limestone. For the simulation without a water layer

on the ground, the evaporation neutrons (100 keV ≤ E < 20 MeV) are moderated by the water in soil and, for example, the flux of those neutrons decreases by about 25 % with a water concentration of 5 % in limestone, as compared to dry limestone. Similarly, the epithermal neutron flux (0.4 eV ≤ E < 100 keV) decreases by a factor of about 2. In contrast, the thermal neutron


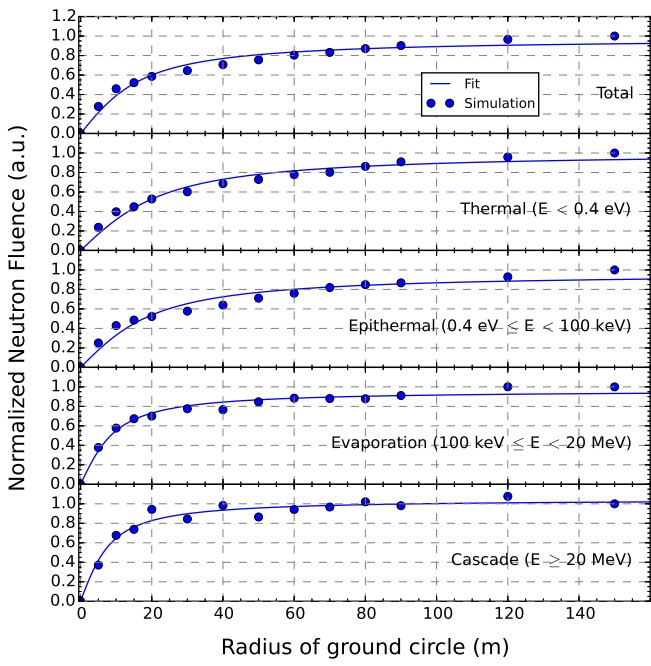

**Figure 11.** Contribution of neutrons with energies in the four neutron energy ranges (thermal: E < 0.4 eV; epithermal: 0.4 eV ≤ E < 100 keV; evaporation: 100 keV ≤ E < 20 MeV; cascade: E ≥ 20 MeV) to the total neutron flux at the scorer position, as function of the radius of the ground area (see Fig. 2). Solid lines: fits to the data using $f(x) = a \arctan(bx)$, with a boundary condition of $f(150\,m) = 1$.

flux (E < 0.4 eV) increases by about 20 % when 5% humidity is added to the soil because the neutrons with higher energies are thermalized by the hydrogen.

Interestingly, when a 5 cm thick water layer is added on top of the limestone, soil moisture has practically no effect on the neutron flux spectra whatever water concentration is considered in the limestone for neutrons with energies greater than or equal to 0.4 eV (epithermal, evaporation, cascade neutrons). It is only the thermal neutrons that are slightly affected by the soil moisture, with the thermal neutron fluence decreasing by about 25 % for a soil moisture of 5 % as compared to dry limestone.

    In contrast, cascade neutrons (E > 20 MeV) are not affected at all by the humidity in the soil, also a 5 cm water layer has no
effect on the cascade neutrons.

## 4   Conclusions

In this study a systematic analysis of the influence of environmental parameters on the neutron flux spectra from secondary CRs at mountain altitudes was performed. For this, Geant4 Monte Carlo calculations were made, and the influence of snow height and soil moisture on neutron flux energy distributions and range of albedo neutrons from soil were investigated.

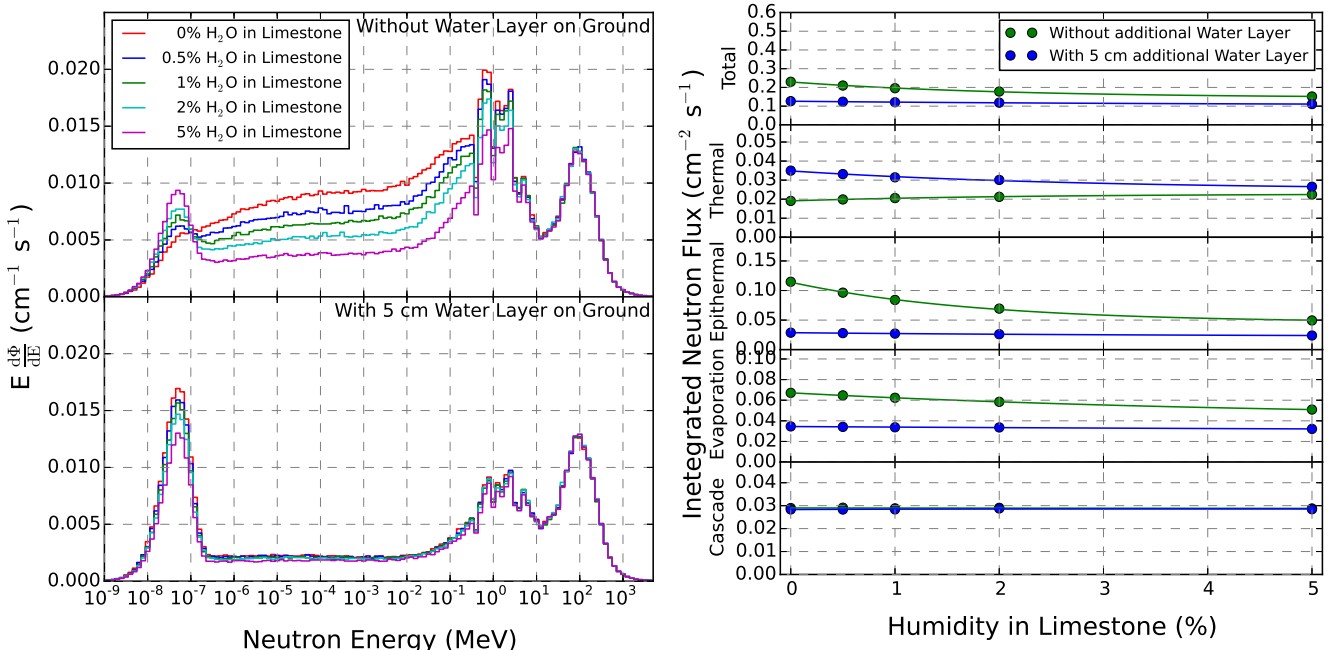

**Figure 12.** Left: Simulated neutron flux spectra with different moisture content in limestone (0, 0.5, 1, 2 and 5 %) without water layer on the ground (top) and with a 5 cm water layer on the ground (corresponding to a snow height of about 20 cm) (bottom). Right: Neutron fluence integrated over the whole energy range ("total") or integrated over the four considered neutron energy ranges (thermal: E < 0.4 eV; epithermal: 0.4 eV ≤ E < 100 keV; evaporation: 100 keV ≤ E < 20 MeV; cascade: E ≥ 20 MeV); solid lines represent fits of the data to $a\,e^{-bx} + c$.

As described in a previous publication, the Geant4 MC simulations were validated experimentally by means of ERBSS measurements of neutrons from secondary CR at UFS Zugspitze, Germany (2650 m a.s.l.) and in the astronomical cupola at the top of the Sphinx observatory on the Jungfraujoch, Switzerland (3585 m a.s.l.) (Brall et al., 2021).

The simulations described in the present paper were done for the UFS at an altitude of 2661.5 m a.s.l. on a limestone ground (which is typical for the Zugspitze region) assuming a horizontal surface. Soil moisture was varied from 0% to 5%, and a SWE

layer with a thickness between 0 cm and 50 cm was added (corresponding to snow heights up to about 200 cm), to investigate the influence of these parameters on the neutron flux spectra. The resulting simulated neutron flux spectra were corrected to account for the slanted area of the Zugspitze mountain as described in Brall et al. (2021), and folded with the ERBSS response functions, to obtain the count rate for each ERBSS [3]He proportional counter. These count rates were then compared to those actually measured with the UFS ERBSS in 2018.

The influence of the snow-depth on neutron fluence shows that SWE affect the neutron fluence, for energies below 20 MeV, strongly up to about 10 cm SWE and for thicker layer of water with more than 20 cm SWE, the effect of neutron absorption by hydrogen saturates, while no more change in the count rates can be recognized in the case of thicker water layers.

The SWE estimate from the measured count rates of the ERBSS at Schneefernerhaus, Zugspitze, provides consistent results for all spheres including the bare detector, except for the 11" and 9,2" spheres, which underestimate the SWE. The mean SWE

values deduced for the winter months (January to March) were between 6.7 and 10.1 cm, and between 2.1 and 3.6 cm for the summer months (July to September). For the summer months also the bare and 2.5" detectors provide somewhat lower values.

Humidity in limestone also has a strong effect on the neutron flux, for neutron energies below 20 MeV. However, the effect of humidity on neutron moderation and absorption only slightly affects thermal neutron flux when a layer of 5 cm of water on top of the limestone is present.

The simulations performed in the present study to investigate the range of albedo neutrons show that a circle with a radius of 40 m contributes about 80 % to the total neutron flux backscattered from the ground to the detector position. Unfortunately, the geometry for the simulation did not allow the use of a point detector, which would have described the real circumstances of the used Bonner spheres more realistically than the chosen scorer disc with a radius of 5 m (to simulate the detector) due to limited computing time. We note that a smaller scorer than the one used in the present study would result in a smaller ground

circle where backscattered neutrons contribute to the detector response.

We conclude that measurements of neutrons from secondary cosmic radiation with a ERBSS system are sensitive to hydrogen in the environment. This holds for the unfolded neutron flux spectrum and for the count rates obtained with the single ERBSS detectors. Specifically, information can be gained on heights of snow cover and its seasonal changes, on soil moisture, but also information on local geometry such as terrain gradients and distance to the origin of albedo neutrons reaching the ERBSS

position. The study performed demonstrates the importance of the measurement of neutron energies, because the influence of the investigated parameters strongly depends on neutron energy. More detailed and quantitative analyses would benefit from an optimized detector design with increased counting statistics.

*Code and data availability.* No code or data available

*Author contributions.* All authors designed the project. TB performed the Monte Carlo simulations and analysed the data. TB and VM

assessed the experimental data of the Bonner Sphere Spectrometer at Zugspitze mountain. All authors discussed and interpreted the results and wrote the manuscript.

*Competing interests.* The authors declare that they have no conflict of interest.

*Acknowledgements.* We would like to thank the staff of the UFS laboratory for their long-term support in the ERBSS measurements. This research received funding from the Bavarian State Ministry of the Environment and Consumer Protection, within the research project "Virtual

Alpine Observatory" under contract number "71_1d-U8729-2013/193-5".



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
