# Peer review of "Assessment of neutrons from secondary cosmic rays at mountain altitudes – Geant4 simulations of environmental parameters including soil moisture and snow cover"

_The Cryosphere, 2021_

## Author Comment (AC1)

Dear Referee,

Thank you very much for your comments and providing helpful suggestions on our manuscript. We paste the reviewer comments in black and our response appears in red. *Quotes from the manuscript are in red italic.*

Page 2, Line 45:

Please check the reference (Simpson ?Meyer and Simpson, 1955).

Thank you, we corrected this reference.

Page 4, Lines 88:

In the 1$^{st}$ step of the two-step calculation, it is necessary to introduce a region that absorbs all radiations (so-called "black-hole" region) below the scoring surface, otherwise some albedo particles are double counted as the source of 2$^{nd}$ step calculation because of multiple scattering between the ground and air. However, such gimmick is not mentioned neither in the text nor in Figs. 1 and 2. If the authors fully transport all radiation in the 1$^{st}$ step calculation, they must perform it again by introducing a black-hole region, or analyze the influence of the multiple scattering on their final results.

The reviewer is right. Our description was too rough, because the method we applied is explained with more details in an earlier paper (Brall et al., 2021) (cite before in line 83 *"The GEANT simulations described in Brall et al. (2021)"*). To clarify and address the critics of the reviewer we added following sentence:
*"To avoid that secondary particles in the simulation are backscattered from the volume below the scoring region and then double counted in the scorer, vacuum has been assumed instead of air below the scoring surface (for details please see Fig. 3 in Brall et al., 2021)."*

Table 1:

Please consider to provide the statistical uncertainties in the table.

We added the statistical uncertainties in the table and also added following sentence to the table caption *"...with one-sigma standard deviation of the Monte Marlo calculation"*

Figure 5:

Please consider to provide the numerical values of a, b, c parameters in this figure. It is beneficial for some readers who want to reproduce the results.

We agree, this will be useful for the reader, thank you. We added the functions of the fitted curves to the figure.

Page 11, Line 214

It is written that "For the summer months when there is no snow at the UFS, the measured values are between 2.7 and 3.6 cm SWE." Assuming the influence of the buildings of the UFS research station was negligible, what is the corresponding moisture content in the limestone as expected from Fig. 12? Then, what is the typical moisture content in the limestone? The reason why I ask these questions is that there is no experimental verification of the simulation results, and this comparison could be a clue for the verification.

The reviewer is quite right that such a comparison could be a clue for verification of our simulations. Unfortunately, we do not have any information on how much moisture might be in the building materials of the research station (concrete always contains some water, water pipes in the research station might also contribute, etc.). Moreover, the actual soil moisture in limestone depends on number of cracks in the stone, etc., which is also unknown to us. We therefore respectfully hesitate to follow the recommendation of the reviewer and do any analysis based on the assumption of zero humidity in the building materials or on the humidity content of local limestone, to avoid any over-interpretation of our results.

To address the reviewer's comment, however, we have changed the following sentence (changes marked in yellow) (page 12, 217-219):

*"Thus, any contribution of the water content in the environment such as the typical 3-10 % water content of concrete, soil moisture and any additional water content in the concrete floor around the detector housing could not be considered in the simulations, due to lack of information."*

*Because we agree with the reviewer in principle, we have added to this sentence: "Because an experimental verification of the simulated results would be a clue for testing the proposed approach, however, BSS measurements of neutron spectra would be desirable in an environment with a defined and well-known humidity in the relevant environmental compartments."*

We have also added to the end of the Conclusions section (page 17, line 298): *"More detailed and quantitative analyses would benefit from an optimized detector design with*

*increased counting statistics, and from detailed BSS measurements in an environment with known hydrogen content in the relevant environmental compartments."*

Reference:

Several authors have already investigated the influence of the soil moisture on the cosmic-ray neutron fluxes using Monte Carlo simulation. For examples, Sato et al. (2006)* and Hubert et al. (2016)** show graphs similar to the upper panel of Fig. 12 in this manuscript. I recommend to cite the earlier works and clarify the difference of this study in the Introduction.

*Sato et al. Radiat. Res. 166, 544 (2006)

**Hubert et al. JGR: Space Phys. 121, 12186 (2016)

Thank you, we cite these references now and write:

*"Sato et al. (2006) and Hubert et al. (2016) did similar calculations on the influence of soil moisture on the neutron fluence, their results are in consistence to the results shown in Fig.12. "*

---

## Author Comment (AC2)

Dear Referee,

Thank you very much for your comments and providing helpful suggestions on our manuscript. We paste the reviewer comments in black and our response appears in red. *Quotes from the manuscript are in red italic.*

The authors Brall et al. present in their paper "Assessment of neutrons from secondary cosmic rays at mountain altitudes (...)" a Monte-Carlo-based study with the motivation to better understand the neutron spectrometer at their experimental site and therefore improve the reliability of the data. The simulations provide key insights into the scaling of the CRN intensity with respect to SWE and partially environmental water. The study seems to some extent however to be incomplete as if the results would not have undergone internal revision. The data comparison to the simulations suggests that the initially simulated soil water contents (<5%) underestimate the actual situation. The range calculations do not extend beyond 150 m although the simulations appear to extend beyond that. Therefore the reviewer is wondering why these iterations have not been carried out if the authors have to conclude that they might not have simulated an appropriate topology.

Scientific value:

- Although the authors focus mainly on their own experimental site their results for the understanding of CRN fluxes are relevant and new. Most calculations in the field have not been carried out using GEANT4. Furthermore the topic of snow height measurements is of upcoming interest.

General:

- The study limits itself to very low hydrogen contents in limestone. The reviewer does not know the particular geology at the Zugspitze, but nevertheless wants to mention that there is alawys a small fraction of chemically bound water, so 0 % is unrealistic, and in general limestone can take more than just 5 % of soil moisture.  In the discussion the authors mention the fact as well as a possible reason for their observations, but the initial assumption that limestone contains no water at all is already not correct. This should have been at least used as a primary assumption for the data being so far off from the simulations for example for the thermal neutrons.

We agree with the reviewer that moisture in the limestone is a critical parameter. As we have stated in our manuscript, we have no information whatsoever on the water content in limestone at the UFS location. Therefore, as a first guess and "worst case scenario" we began our simulations with the assumption of pure (dry) limestone. To investigate the theoretical influence of soil moisture in more detail, we then systematically increased the water concentration in limestone up to 5% (Fig. 12). The real water content in the soil material close to the measurement position, as well as

that in the structural material of the measurement platform and the building material of the UFS is also unknown (note that concrete always contains some water, water pipes in the research station might also contribute, etc). Therefore, any assumption on the water content in soil and building structure would be highly speculative. We therefore respectfully hesitate to follow the recommendation of the reviewer and discuss the soil moisture content of local limestone in more detail, to avoid any over-interpretation of our results.

To address the reviewer's comment, however, we have changed the following sentence (changes marked in yellow) (page 12, 217-219):

*"Thus, any contribution of the water content in the environment such as the typical 3-10 % water content of concrete, soil moisture and any additional water content in the concrete floor around the detector housing could not be considered in the simulations, due to lack of information."*

Because we agree with the reviewer in principle, we have added to this sentence: *"Because an experimental verification of the simulated results would be a clue for testing the proposed approach, however, BSS measurements of neutron spectra would be desirable in an environment with a defined and well-known moisture in the relevant environmental compartments."*

We have also added to the end of the Conclusions section (page 17, line 298): *"More detailed and quantitative analyses would benefit from an optimized detector design with increased counting statistics, and from detailed BSS measurements in an environment with known hydrogen content in the relevant environmental compartments."*

- the US Standard Atmosphere 1976 does not contain argon. The thermal capture cross section of argon however is not negligible for thermal neutron studies. Furthermore there seems to be no air humidity involved, which is also relevant for the overall neutron flux.

The U.S. Standard Atmosphere 1976 does contain Argon (see for example https://www.google.com/search?client=firefox-b-d&q=us+standard+atmosphere+1976+table). Thus, argon was considered in our study. The chemical composition is also given in a previous publication (Brall et al. 2021).

The reviewer is right that we used - as mentioned in our paper – only dry air, because we believed that air as a pool of environmental hydrogen is less important in high mountains than, for example, soil moisture or snow cover. However, to address the comment of the reviewer we have added to the discussion section: *"Note that other pools of hydrogen in the environment, such as air humidity or vegetation, were not investigated in the present study."*

- The fact that the high-energy neutron flux changes in the slanted geometry as well, it seems that the 'flux reduction' is an entirely cos(angle) geometrical effect due to inclining the flat scorer. Or do you state that a volumetric detector would still measure that difference?

We indeed used a volumetric detector and not a flat scorer.

  - why have the homogeneous calculations not been carried out with periodic boundary conditions?

We thank the referee for this useful comment. This would have been indeed an option if CPU time was an issue. However, we were able to perform our simulations with the computer capacity available.

- There are no statements about statistical uncertainties. This should be corrected.

The reviewer is right, thank you for the remark. We added the following sentence to section 2.1: *"The number of primary particles was chosen such that the statistical uncertainties of the scored neutrons was less than 3 % for total neutron flux (and less than 6 % per energy region of interest)."*

- There is a conceptual problem in the distance contribution calculations: Firstly, neutrons scatter in the atmosphere - soil interface several times. Placing a circle in a vacuum is not equal to analyzing the distance contribution in an infinite scenario. Neutrons might originate from a distance within 5 m scatter around and then reach the scorer. If you replace everything around the radius of interest with vacuum you inhibit this process. That may be less relevant for 100 m or 200 m but for small radii it is definitely a misrepresentation. Secondly, as the cited literature values provide significantly larger origin distance values for the 86 % quantile it seems incorrect that the authors simply stop their analysis at 150 m and normalize that to 1. It is not clear how much the flux would increase beyond 150 m and the provided data points already by optical inspection clearly show a trend which indicates that 150 m is definitely not 100 %. Therefore the authors should extend their distance calculation at least by a factor of 2 or 3 and/or use an 'infinite' soil as a reference value for 100%.

We agree with the referee that the chosen approach may have been too simplistic, in particular for short distances, i.e., for radii which are of similar size as the scorer radius. However, for distances which are large as compared to the scorer radius this

might not be critical. We also agree that a radius of 150 m may have been too small, although the effect of increasing the radius further would probably not change the final outcome by more than a few percent.

Unfortunately, due to closure of the project, we have no possibility to investigate this issue further. Therefore, to address the critics of the referee, we have decided to remove this whole section from the paper.

Misc:

- in section "3 Results and Discussion" there is a 3.1 with subsequent subsections but no 3.2

Thank you very much for spotting this. We have corrected the numbers of the subsections.

- the differential neutron flux is stated in units of per cm^2 not per cm as most plots show

Thank you, this is now corrected in the plots.

- why do the authors set variables like in the polynomials in italic whereas others like the energy E is not?

Thank you very much. We have now set all variables in italics.

- the term intensity is used in the introduction but not in the rest of the manuscript. In the later sections the terms flux, fluence and fluence rate (which equals flux) are used in a maybe not so clearly distinguishable way.

Thank you very much. We checked the manuscript for consistent terminology. However, when we use the word "intensity" it is used in more general terms as the word "flux" and this should be okay.

- the plots are tiny (!). The reviewer would be surprised if the authors can read values properly from the graphs when printed on A4.

Thank you very much. We propose to wait for advice from the editor whether any change in the styles of the figure is necessary.

References:

- some previous works which are relevant to this study are missing, that might be seen optional, however, the authors seem to be quite keen on citing their own works, so a more balanced reference list would be advised. The following works can be taken into account:

- Hendrick 1966 10.1103/PhysRev.145.1023 - probably the first source to discuss the neutron flux relation with respect to soil moisture

- Kodama 1980 10.1016/0165-232x(80)90036-1 - first application of CR neutrons to measure snow

- Schattan 2017 10.1002/2016WR020234 and Schattan 2019 10.1029/2019WR025647 - above-ground neutron measurements of snow water

- Desilets 2010 10.1029/2009WR008726, Franz 2013 10.5194/hess-17-453-2013, Koehli 2021 10.3389/frwa.2020.544847 above-ground neutron intensity relation

- Desilets 2013 10.1002/wrcr.20187 followed by the mentioned Koehli 2015 for the CR footprint

- Sato 2006 10.1667/RR0610.1, Sato 2015 10.1371/journal.pone.0144679 and Sato 2016 10.1371/journal.pone.0160390 for CRN flux above the ground (PHITS)

- maybe Nesterenok 2013 10.1016/j.nimb.2012.11.005 for CRN flux using GEANT4

Thank you very much. We refer to these references in the introduction of the revised version.

Abstract:

- L4: "To investigate the impact of these parameters, Geant4 Monte Carlo simulations were carried out." - It should be mentioned that in particular 'these parameters' are snow cover.

"To investigate the impact of these parameters, in particular snow cover, Geant4 Monte Carlo simulations were carried out."

Introduction:

- L30: "due to its large elastic scattering cross section" - and equal mass of projectile and target

Thank you very much for pointing this out. We now write: *"due to its large elastic scattering cross section and equal mass of projectile and target."*

- L43: as a sidenote: it was not just Hess, in the years 1910-1914 a series of balloon flight radiation measurements were carried out by Wulf, Gockel and Hess

Thank you very much for pointing this out. We now write *"Inspired from earlier work of Wulf and Gockel (1,2,3)..."*

And add following references: (1)Th. Wulf, Phys. Z. 10, 997 (1909), (2)Th. Wulf, Phys. Z. 11, 811 (1910), (3)A. Gockel, Phys. Z. 12, 595 (1911)

- L45: It is unclear which reference the author means, but the IGY was presented as early as 1953 (10.1103/PhysRev.90.934)

Thank you very much. We have added the reference to the reference list.

Simulation Geometries:

- L113: Sentence incomplete

Thank you, corrected, there should be a new line after "... function of distance".

Difference between horizontal and slanted soil:

- Fig. 3 might be omitted. The influence of a snow water layer is shown more clearly in Fig. 4 and the slanted geometry has no other effect than changing the rates.

To be honest we would prefer keeping the figure, because the effect of slanted geometry is more descriptive and supports the information in table 1.

Influence of snow height on neutron flux spectra:

- L148: There is a vagabond "Simulations" which may in combination with the later appearing "Interpretation in terms of thickness of water layer" be meant as a section. As your section layout in 3. is anyway not carried out properly, see above, subsubsection might be introduced here.

Thank you for pointing this out. We have deleted "Simulation" and rearranged our section layout accordingly.

- L159: "In contrast, for high-energy cascade neutrons, this decrease is small and amounts only to about a few percent" - shouldn't the high energy part be nearly entirely be oriented in forward direction? At such high kinetic energies the backward direction is heavily suppressed - which might be represented by the data as well.

We agree with the referee that forward scattering of high energy neutrons is dominating and backward scattering is suppressed. Nevertheless, there are few high energy neutrons backscattered from soil, which are then moderated by the water layer. This is reflected in figure 5.

- the results presented in Fig. 5 are indeed quite relevant to the field. They should be discussed in more detail. However, it would be more appropriate to plot the results from for example the 5 % soil moisture runs as 0% water in soil and air constitutes an extreme and unrealistic scenario.

To address the comment of the reviewer, we have now added details on the fit parameters of the curves to the figure.

Unfortunately, as discussed already above, we do not have any information on the limestone moisture and the water content of structural material of the research station, close to the vicinity of our BSS. Therefore, we hesitate discussing any (necessarily) speculative soil moisture value. As also indicated above, we have addressed the issue in the revised manuscript and added to the conclusion section that more research should be done in an environment where the water content of the environment is precisely known.

- L210: Schattan 2017 and 2019 showed that homogeneous and inhomogeneous snow distributions do not show the same response. That means that for patchy snow cover the SWE values you state might correspond to an effective SWE but not to the actual one.

Thank you very much for pointing this out. We have now added the following sentences: *"Because we had no detailed information on the type of snow present at the measurement times (density etc.), all simulations were based on SWE values. We note,*

*however, that Schattan (2017, 2019) have shown that homogeneous and inhomogeneous snow distributions show a different influence on the neutron field."*

Humidity of Limestone:

- it should be more correctly called moisture instead of humidity

Thank you, we have changed the wording accordingly throughout the manuscript ("moisture" instead of "humidity")

- Fig. 12 (right) the y-axis range should be reduced in order to stretch the scaling of the presented values.

Done

- It would be interesting to see a comparison of the neutron flux results presented here to those already published in literature. At least the reviewer is aware of the fact that Sato 2006 already presented analytical function for the CR intensity depending on the soil water fraction.

Thank you very much for pointing this out. Sato et al. 2006 and Hubert et al. 2016 is now cited as follows:

*"Sato et al. (2006) and Hubert et al. (2016) did similar calculations on the influence of soil moisture on the neutron fluence, and their results are in line with those shown in Fig. 12."*

- It can be expected that the thermal neutrons do not only show this 'anomaly' for their scaling with respect to SWE but likewise to soil moisture as well. The reviewer however is aware of the fact that the authors might not have conducted such runs as well.

The reviewer is quite right. We have therefore added the following sentence: *"It may well be that if the soil moisture was increased further in the simulations, a decrease in thermal neutron flux might have been observed (similar to the behavior of the thermal neutron flux as a function of SWE (Figs. 5 and 7))."*

Conclusions:

- L262: "systematic analysis of the influence of environmental parameters" - as pointed out above, the main focus is set on snow.

The reviewer is correct. For this reason, we had explicitly mentioned in the following sentence snow height. We hope this is ok.

- L280: as stated above, those SWE values might not be too reliable considering the arguments brought up.

We agree with the referee and have therefore added: *"Note that these results should be taken with care because considerable uncertainties remain."*

- L299: The issue of computing time is brought up here out of the blue. How many neutrons do you simulate and how long does it take?

We have decided to delete the whole section 3.13 (see above). Therefore, the corresponding paragraph in the conclusion, to which the referee is referring, is also deleted.

---

## Editor Decision (ED1)

2021-08-16
Submission tc-2021-152

Dear Dr. Brall thank you for your submission to be considered for publication in The Cryosphere. Your manuscript entitled '*Assessment of neutrons from secondary cosmic rays at mountain altitudes – Geant4 simulations of environmental parameters including soil moisture and snow cover*' I thought was very interesting. I am aware of the use of cosmic ray radiation work from a SWE perspective, therefore aware of the sensitivity to moisture etc. I thought quite innovative to use Monte Carlo simulations of neutrons for alpine regions.

From my own reading of the comments to the reviewers, I think the authors did a great job answering the questions and modify portions of the paper accordingly. Therefore, I accept the paper to be published following those comments.

Thank you for submitting your work to The Cryosphere.

Regards,
Prof. Dr. Alexandre Langlois
Associate editor, *The Cryosphere*